# Metabolome and Transcriptome Analyses Reveal the Regulatory Mechanisms of Photosynthesis in Developing *Ginkgo biloba* Leaves

**DOI:** 10.3390/ijms22052601

**Published:** 2021-03-05

**Authors:** Ying Guo, Tongli Wang, Fang-Fang Fu, Yousry A. El-Kassaby, Guibin Wang

**Affiliations:** 1Co-Innovation Centre for Sustainable Forestry in Southern China, Nanjing Forestry University, Nanjing 210037, China; yingguo@njfu.edu.cn (Y.G.); fffu@njfu.edu.cn (F.-F.F.); 2Department of Forest and Conservation Sciences, Faculty of Forestry, The University of British Columbia, Vancouver, BC V6T 1Z4, Canada; tongli.wang@ubc.ca; 3Forestry College, Nanjing Forestry University, Nanjing 210037, China

**Keywords:** ginkgo, photosynthesis, carbon metabolism, transcription factors, gene co-expression network

## Abstract

Ginkgo (*Ginkgo biloba* L.) is a deciduous tree species with high timber, medicinal, ecological, ornamental, and scientific values, and is widely cultivated worldwide. However, for such an important tree species, the regulatory mechanisms involved in the photosynthesis of developing leaves remain largely unknown. Here, we observed variations in light response curves (LRCs) and photosynthetic parameters (photosynthetic capacity (P_nmax_) and dark respiration rate (R_d_)) of leaves across different developmental stages. We found the divergence in the abundance of compounds (such as 3-phospho-d-glyceroyl phosphate, sedoheptulose-1,7-bisphosphate, and malate) involved in photosynthetic carbon metabolism. Additionally, a co-expression network was constructed to reveal 242 correlations between transcription factors (TFs) and photosynthesis-related genes (*p* < 0.05, |r| > 0.8). We found that the genes involved in the photosynthetic light reaction pathway were regulated by multiple TFs, such as bHLH, WRKY, ARF, IDD, and TFIIIA. Our analysis allowed the identification of candidate genes that most likely regulate photosynthesis, primary carbon metabolism, and plant development and as such, provide a theoretical basis for improving the photosynthetic capacity and yield of ginkgo trees.

## 1. Introduction

Photosynthesis comprises temporal and spatial biochemical reactions that act in concert to convert light energy into chemical energy, representing the primary mechanism of dry matter production in plants [1]. The main function of the light-dependent reactions is the absorption and conversion of light to provide assimilatory power for the light-independent reaction, which in turn assimilates CO_2_ into carbohydrates [2]. For trees, photosynthesis improvement could increase carbon fixation and the production of biomaterials such as wood and fiber [3]. Carbon fixation is integrated over the entire growing season, so even small increases in the rate of photosynthesis can translate into yield increase [4].

The leaf growth process follows a complex cellular order, including proliferation, expansion, and maturation of cells, to reach its mature shape and size [5]. During leaf development, not only does leaf morphology change significantly, but also the structure and function of the photosynthetic apparatus [6]. With leaf expansion, the chlorophyll content, stomatal conductance, photosynthetic rate, carboxylation efficiency, related enzyme activities, and CO_2_ assimilation rate also increase [7]. For most higher plants, the light-saturated net photosynthetic rate on a leaf area basis peaks at or slightly before full leaf area expansion [8]. The photosynthetic capacity decreases with leaf senescence, which was mainly associated with the loss of ribulose-1,5-bisphosphate carboxylase [9]. Hence, in view of the difference in photosynthetic rate across leaf developmental stages, improvement of leaves’ photosynthetic efficiency can be achieved via different strategies.

Several studies have recently explored the gene functions involved in photosynthesis, aiming to discover new approaches to improve photosynthesis and increase plant productivity [10,11]. Transcription is the initial step for a gene to be selectively expressed and regulated. Transcription factors (TFs) are key regulators, which regulate the expression of target genes by binding to their cis-regulatory elements, thus affecting the corresponding plant phenotypes [12]. The growing identification of interactions between TFs and photosynthesis-related genes through transcriptome sequencing has facilitated the elucidation of the photosynthesis regulation mechanism. For example, a study on *Triticum aestivum* has indicated that the nuclear factor Y (NF-Y) transcription factor was involved in the positive regulation of a number of photosynthesis genes. Thus, transgenic wheat lines overexpressing the NF-Y had a significant increase in their leaf chlorophyll content and photosynthesis rate [13]. In *Arabidopsis thaliana*, Golden2-like (GLK) transcription factors help in regulating and synchronizing the expression of a suite of nuclear photosynthetic genes and thus act to optimize photosynthetic capacity [14]. Additionally, a gene co-expression network for *Populus tomentosa* was constructed to elucidate the mechanisms by which underlying genetic variation in TFs affects complex photosynthesis-related traits [3].

Ginkgo (*Ginkgo biloba* L.) is a deciduous tree species and is one of the most ancient living gymnosperms. Due to its high timber, medicinal, ecological, ornamental, and scientific values, the species is widely cultivated worldwide [15,16,17]. To date, considerable efforts have been made to explore the photosynthetic functions of the ginkgo leaf at the physiological level. For instance, previous studies have analyzed the effects of gender, natural senescence, and seasonal variations on ginkgo photosynthetic functions [18,19,20]. Further, the photosynthetic responses of ginkgo to ozone exposure, high and low irradiance, and soil water stress were also studied [17,21,22]. To increase and/or manipulate ginkgo photosynthesis, we must first understand the molecular mechanisms during leaf development. However, to the best of our knowledge, for such an important tree species, the regulatory mechanisms involved in the photosynthesis of developing leaves remain largely unknown.

In order to fill this knowledge gap, we generated an omics dataset (transcriptomics and metabolomics) for ginkgo leaves at four different developmental stages to identify several photosynthesis-related metabolites and target genes that might be subjected to further studies in order to increase the species photosynthesis efficiency. Here, we sought to: (1) investigate changes in light response curves and photosynthetic carbon metabolism during ginkgo leaf development, (2) identify differentially expressed genes and transcriptional factors involved in photosynthesis, and (3) construct associated transcriptional regulatory networks. We believe that understanding the photosynthesis physiological, molecular, and related processes is essential for sustaining or even improving ginkgo productivity.

## 2. Results

### 2.1. Variations in Leaf Morphology and Photosynthesis-Related Parameters

To understand the variation of photosynthesis-related traits among the different ginkgo leaf developmental stages, leaves’ morphological parameters were evaluated and light response models were determined. Generally, leaf growth rate across the four developmental stages (T_1_–T_4_) was in a stationary-state (Figure 1A,B); however, significant differences (*p* < 0.05, N = 27) in leaf width (Figure 1C) and leaf area (Figure 1D) were observed.

Photosynthetic rate (P_n_) data were fitted against photo-synthetically active radiation (PAR) using the modified rectangular hyperbola model in Equation (1) (R^2^ range: 0.93 to 0.99) (Figure 2A). The P_n_ trajectory increased sharply with the increase in PAR when PAR was low, followed by a slow increase until reaching maximum P_n_. Light response curves (LRCs) differed among the four leaf-developmental stages, and this difference was small when PAR was low (PAR < 500 μmol∙m^−2^∙s^−1^), and substantially increased with increasing PAR.

For photosynthetic light response parameters, photosynthetic capacity (P_nmax_) gradually increased with increasing leaf development, and P_nmax_ reached its maximum value (12.85 μmol m^−2^∙s^−1^) at T_4_ (Figure 2B). Dark respiration rate (R_d_) and light compensation point (I_c_) initially increased, reaching their maximum at T_3_, and then slightly decreased at T4 (Figure 2C,D). Nevertheless, the maximum light saturation point (I_sat_) appeared at the early developmental stage (T_1_, I_sat_ = 2185 μmol∙m^−2^∙s^−1^) (Figure 2E).

### 2.2. Variation in Metabolic Levels

N and C metabolic levels were tightly linked to photosynthesis-related biological processes. We found total C and N contents at the different leaf-developmental stages were relatively stable. While total C content gradually increased with leaf-developmental stages, these differences were not significant (Figure 3A). Conversely, total N content gradually decreased with leaf-developmental stages, and only significant differences (*p* < 0.05) were observed between T_1_ and the other three development stages (Figure 3B).

Based on hierarchical clustering analysis, the profiles of the 32 photosynthesis-related metabolites were clustered into four major groups (Figure 3C). Comparisons of the differences in the abundance of photosynthesis-related metabolites among the four leaf-developmental stages (Appendix A), it indicated that T_1_ was associated with metabolites involved in carbon fixation in photosynthetic organisms. For instance, 3-phospho-D-glyceroyl phosphate and glycerone phosphate in the first group were significantly higher (*p* < 0.05) than that of T_4_. Yet, these differences were not observed in the second metabolites group, which also participated in carbon fixation in photosynthetic organisms. Changes in tricarboxylic acid (TCA) cycle substrates, acetyl-CoA, succinyl-CoA, citrate, and thiamin di-phosphate, were clustered in group 3. Their abundances gradually decreased during leaf development. Changes in metabolites involved in the pentose phosphate pathway (OPP) were clustered in the fourth group, such as malate, gluconolactone, d-glucono-1, 5-lactone, 3-phosphoglycer ate, d-glucose, d-glycerate, and d-ribose 5-phosphate, and their abundance gradually increased with leaf development.

### 2.3. Transcriptome Reprogramming

mRNAs from the four leaf-developmental stages (T_1_–T_4_) were sequenced, and approximately 700 million clean reads from the 12 cDNA libraries were identified. They were used to evaluate the leaf-developmental dynamics. Based on gene expression, principal component analysis (PCA) was used to estimate the distance relationship among the 12 samples (Figure 4A). PCA score plots showed a clear separation between samples from analyzed leaf-developmental stages, and they were separated along PC1 with a left-to-right trend following the developmental stages (T_1_ to T_4_). Further, the heat map of gene expression obtained from the hierarchical cluster analysis showed that the 12 studied samples formed four groups, and samples of each developmental stage were classified within their respective group, with a reasonably high correlation (Figure 4B). Taking the T_4_ sample as the control group, the number of differentially expressed genes (DEGs) identified between T_1_ and T_4_ was the largest (N = 4991) (Figure 4C), and most of them were down-regulated; while the least DEGs were found between T_3_ and T_4_ samples (N = 204), and most of them were up-regulated DEGs (Figure 4D). This analysis showed that ginkgo leaves had undergone a drastic transcriptional reprogramming along the leaf-developmental processes.

### 2.4. Temporal Dynamics of DEGs Expression

Co-expression profiles of the DEG sets across the four leaf-developmental stages showed that thousands of genes were classified in 20 different oscillating patterns (Appendix A). Of these, seven were identified with significant (*p* < 0.05) temporal expression patterns (Figure 5). Additionally, we highlighted the five most over-represented KEGG pathways for each profile. The 1056 DEGs from the 0, 2, and 7 profiles (see Appendix A) presented a downward trend, with high expression at T_1_ and decreasing expression towards the T_4_ developmental stage (Figure 5A–C). Notably, in profile 2, multiple genes involved in photosynthesis were found showing significantly higher transcript abundance at T_1_ (Figure 5B). Further, the 1330 DEGs from the 12, 17, 18, and 19 profiles (see Appendix A) presented an opposite trend, with low expression at T_1_ and increasing expression towards T_4_, which were mainly enriched in the pathways of protein transport in endoplasmic reticulum, mRNA surveillance, amino sugar, and nucleotide sugar metabolism, and glutathione metabolism (Figure 5D–G).

### 2.5. Gene Co-Expression Network Construction

To uncover the potential biological regulatory networks modulating the expression of the photosynthesis-related genes, we identified 45 DEGs involved in the biological process of photosynthesis as target genes. According to their molecular functions and subcellular locations, these photosynthetic genes were grouped into three categories: membrane-based photosystem components, electron transport genes, and photosynthesis regulating genes (Appendix A). Further, we defined 28 candidate regulators (transcription factors, TFs) by promoter analysis (Appendix A). Following this, a network based on the co-expression analysis was constructed to reveal 242 correlations between TFs and photosynthesis-related genes (*p* < 0.05, |r| > 0.8) (Figure 6). We found that ARF (*Gb_39786*), TFIIIA (*Gb_35789*), WRKY (*Gb_32055*), and PHR (*Gb_26822*) were correlated with most of the photosynthetic genes (degree ≥ 25).

Interestingly, we found seven genes in the photosynthesis pathway (ko00195) were regulated by several TFs (Appendix A). For example, photosystem I reaction center subunit III (PSAF) and photosystem I subunit O (PSAO) were correlated with most TFs, which were positively and negatively regulated by S1FA and ARF, PHR, IDD, TFIIIA, WRKY, bHLH, and ATHB, respectively. Hence, we constructed a TFs regulating network of photosynthetic light reactions to better understand this pathway (Figure 7).

## 3. Discussion

Recent studies indicated the existence of a positive correlation between photosynthesis and biomass [23,24]. For trees, improved photosynthesis has the potential to increase carbon fixation and biomaterials production [3]. While it is well known that the relationship between light and photosynthesis reflects physiological and biochemical processes [25], but the molecular mechanisms of this process in ginkgo developing leaves remain largely unknown. This study sought to investigate changes in light response curves (LRCs) and primary carbon metabolism during ginkgo leaf development. Additionally, we identified molecular differences among different leaf-developmental stages and constructed a schematic model of transcription factors (TFs) involved in regulating the photosynthesis pathway. These results could deepen our understanding of photosynthesis and contribute to pinpointing potential ways to improving photosynthetic efficiency.

### 3.1. Changes of Photosynthetic Performance of Leaves during Development

Leaf development is considered to be one of the chief factors affecting plant photosynthesis and productivity [26]. This study used fully expanded young (T_1_) to medium-aged (T_4_) leaves collected from May to August, which corresponded to the period when certain leaf traits (e.g., leaf length, leaf width, and leaf area) were relatively stable (Figure 1), but secondary thickening and lignification of leaves continued throughout the leaf life-span [27]. Additionally, we observed differences in light response curves (LRCs) of leaves at the different developmental stages (Figure 2). LRC is a useful tool in plant physiology to assess the photosynthetic performance of plants [28], and the modified rectangular hyperbolic model could be used to describe the photosynthetic capacity and efficiency [29]. In common with most plants [9,30], the photosynthetic capacity (P_nmax_) increases during leaf development (before senescence), reaching a maximum at T_4_. Moreover, we found that variation in dark respiration rate (R_d_) scaled proportionally with variation in P_nmax_, probably since high P_nmax_ required large complements of enzymes and metabolites, which were costly to maintain and needed periodic re-synthesis [31]. Previous studies found a lower sensitivity to photoinhibition of younger leaves [32,33], similarly, the maximum light saturation point (I_sat_) was observed at T_1_. One of the most common causes for reductions in P_nmax_ and R_d_ at T_1_ was chill-induced photoinhibition, which was most likely to occur during late spring/early summer [34]. Meanwhile, we suggested that the increasing leaf thickness could be the main reason for the improvement of photosynthetic performance at T_4_. A study on potato (*Solanum tuberosum*) showed that the relatively thicker the palisade tissue and the resulting higher chlorophyll content per unit of leaf area allowed reaching higher overall photosynthetic rate [35].

### 3.2. Photosynthetic Carbon Metabolism

Metabolism in leaves is dominated by photosynthesis [36]. The primary pathway of carbon fixation in ginkgo (C3 plant) leaves is the Calvin cycle located in the chloroplast stroma [37]. Adenosine triphosphate (ATP) and nicotinamide adenine dinucleotide phosphate (NADPH) are consumed in the reduction of CO_2_ to triose phosphate and the continuous regeneration of ribulose-1,5-bisphosphate (RuBP). The maximum abundance of metabolites involved in this pathway, such as 3-phospho-D-glyceroyl phosphate and sedoheptulose-1,7-bisphosphate, were observed in young leaves (Figure 3C). Previous research indicated that sedoheptulose-1,7-bisphosphatase (SBPase) had the second greatest control coefficient over carbon assimilation [38]; however, in young tobacco leaves, when SBPase activity decreased, the rate of photosynthetic carbon assimilation was unaffected, or even increased [39]. Photosynthetic carbon was used either for growth and biosynthetic processes, involving glycolysis and tricarboxylic acid (TCA) cycle [40]. We found acetyl-CoA, succinyl-CoA, citrate, and thiamin diphosphate abundance were involved in growth and biosynthetic processes gradually decreased, which were used not only in respiration but also in the biosynthesis of other metabolites (e.g., amino acids) [41]. Therefore, we hypothesized that the abundance of the above intermediates (acetyl-CoA, succinyl-CoA, citrate, and thiamin diphosphate) was decreased due to the increase of other metabolites’ biosynthesis during leaf development [42]. Conversely, malate abundance increased gradually, which was consistent with P_nmax_ and R_d_. Previous studies additionally provided strong evidence that malate accumulation could significantly improve stomatal conductance and photosynthetic rate [43]. These results demonstrated that variation in photosynthetic capacity might be a strategy for maintaining the primary carbon metabolism to regulate growth and development [44].

### 3.3. Transcriptional Regulatory Networks in Photosynthesis Gene Expression

To carry out photosynthesis, plants require a large cohort of genes to encode proteins that capture light energy, store energy in carbohydrates and build the subcellular structures required to facilitate this energy capture [1]. Not all cells that are exposed to light develop photosynthetically active chloroplasts [45], hence, photosynthesis-driven gene expression is considered to be strictly controlled by plant development. Interestingly, in the co-expression profiles of the DEG sets across the four leaf-developmental stages, we found that the transcriptional expression of a group of genes involved in photosynthesis increased significantly in young leaves (Figure 5B). As the “switch” of gene expression, the status of a gene’s promoter largely determines when and how intensively it will be transcribed. Notably, this control function of promoters is mostly regulated by transcription factors (TFs) [3]. Here, we constructed a gene co-expression network to reveal the regulatory relationship between TFs and photosynthesis-related genes (differentially expressed target genes) (Figure 6).

We found a significant negative correlation (*p* < 0.05, r < −0.8) between bHLH (*Gb_29528*) and 15 target genes. The bHLH family genes, which were generally considered to be negative regulators of photosynthesis gene expression, bound the G-box motif (CACGTG) found preferentially in the promoters of photosynthesis genes to block transcription [46]. Additionally, bHLH might play a role in controlling the abaxial-adaxial polarity of the maize leaf [47] and stomatal development of *Arabidopsis* leaf [48]. ARF (*Gb_39786*) and IDD (*Gb_21530*) were core TFs in the regulatory network of this study, but to be also considered to be related to stomatal development [49,50]. TFIIIA is widely regarded as the archetypal zinc finger protein, which interacts with the photosystem II (PSII) core complex [51]. In this study, we identified a gene encoding TFIIIA (*Gb_35789*), which had a significant correlation with the multiple target genes. Moreover, two genes encoding WRKY (*Gb_32055* and *Gb_01873*) were found to regulate multiple target genes. WRKY proteins with phosphorylated binding sites could form complexes with other proteins, thus indirectly participate in many cellular events [52].

In the first stage of photosynthesis, the light-dependent reactions provide the reducing power for carbon fixation, and this process involves photosystem I (PSI), PSII, the ATP synthase complex, and the Cyt b_6_f supercomplex [53]. Interestingly, we found that the genes encoding these complexes were regulated by WRKY (Figure 7). ARF, IDD, and TFIIIA negatively regulated the gene expression of *GbPSAF*, *GbPSAO*, and *GbPSBY*, while S1FA had a positive regulatory effect (Appendix A). Additionally, we found *GbATPA* was positively regulated by DOF (*Gb_07177*) and *SPL* (*Gb_22010*). *DOF* could function as transcriptional activators or repressors of light-regulated gene expression [54], and *SPL* played a crucial role in maize growth and development [55]. The gene co-expression network constructed in this study elucidated the regulatory mechanism of TFs on photosynthesis-related genes, thus improving our understanding of photosynthesis and identifying potential ways to improve photosynthetic efficiency.

## 4. Materials and Methods

### 4.1. Plant Materials

In March 2018, scions were collected from a 30-year-old female ginkgo and grafted onto three-year-old ginkgo seeding (N = 60 trees), whose tree height was controlled at 1.4–1.8 m and ground diameter at 1.5–2.0 cm. Both rootstock and scion originated from the Dafozhi variety. The grafts were planted in a test site located in the Pizhou Ginkgo Seed Base (118.05° E, 34.30° N) in a randomized complete block design with three blocks, with each block harboring 20 Ginkgo clones planted at 40 × 60 cm spacing. The test site has a mean annual temperature of 16 °C, mean annual precipitation of 671 mm, and its soil attributes are as follows: total carbon content of 1.15 g/kg, total nitrogen content of 0.44 g/kg, and total phosphorus content of 0.40 g/kg, and 7.58 pH.

### 4.2. Observations of Leaf Morphology and Photosynthetic Parameters

In 2019, leaf morphology and photosynthetic parameters of the studied samples were determined monthly between leaf expansion (May, after the majority of leaves’ expansion) and leaf “commercial” ripening (August, before autumnal senescence). Measurements were made between 9:00–11:30 a.m. on sunny days in the middle of each month to represent four temporal leaf developmental stages (i.e., May, June, July, and August designated as T_1_–T_4_, respectively). Leaf morphology attributes (length, width, and area) were measured from a randomly selected tree within each block, and then three fully grown leaves from the top, middle, and bottom of each tree’s crown were randomly scanned (YMJ-B Area Meter, Top Cloud-ARGI Inc., Hangzhou, China).

Additionally, photosynthetic rates were measured using a CIRAS-3 portable photosynthetic system (PP Systems, Amesbury, MA, USA) for these selected trees conducted on two healthy and fully-grown leaves in the middle of these trees’ crown. The selected leaves were marked for the consecutive measurements. The photosynthetic rates were measured under different photosynthetically active radiations (PAR) (2000, 1700, 1400, 1200, 1000, 800, 600, 400, 300, 200, 150, 100, 50, and 0 μmol∙m^−2^∙s^−1^) for three times per each marked leaf, so six measurements (2 leaves × 3 times) were made for each tree. During measurement, the atmospheric CO_2_ concentration was maintained at 400 ± 20 μmol∙mol^−1^, relative humidity at 60 ± 4%, and the leaf chamber’s temperature was consistent with the ambient temperature. For each PAR, measurement time was controlled to 3 min and the photosynthetic parameters were recorded automatically by the instrument. Each specific light response curve (LRC) was fitted using the modified rectangular hyperbolic model [56], expressed as follows:(1)$$P_{n}\;=\alpha \times \frac{1-\beta \times I}{1+\gamma \times I}\times I-R_{d}$$
where P_n_ is the net photosynthetic rate, α is the apparent quantum yield, β and γ are modified coefficients, R_d_ is the dark respiration rate, and I is PAR. Four relative photosynthetic parameters; namely, photosynthetic capacity (P_nmax_), dark respiration rate (R_d_), light compensation point (I_c_), and light saturation point (I_sat_), were obtained using the http://photosynthetic.sinaapp.com accessed on 20 May 2020.

### 4.3. Carbon and Nitrogen Determination

A single Ginkgo tree was randomly selected in each block to provide leaf samples for total carbon (C) and nitrogen (N) determination. Each sampled tree was represented by three-crown positions (top, middle, and bottom) to provide a single complete and healthy leaf per each crown position. Leaf samples were collected monthly, dried (70 °C, 48 h), crushed, and sieved through a 100-mesh sieve. Dry leaf powder (50 mg) packed with foil was placed in an element analyzer (vario MACRO cube, Elementar, Heraeus, Germany) to determine the Ginkgo leaf samples C and N contents. One-way analysis of variance (ANOVA) was carried out for C and N contents using SPSS software (Version 22.0, IBM, Armonk, NY, USA) and significant differences were calculated using the least significant difference (LSD) test at *p* < 0.05.

### 4.4. Metabolomics Analysis

Following Guo et al. [57], leaf samples were collected and their metabolites were extracted and detected. Briefly, a single clone was randomly selected across the three blocks (i.e., three biological replications), and three leaf samples were collected from the top, middle, and bottom crown on a clear day in the middle of each month (May to August). Collected leaves were immediately placed in liquid nitrogen, freeze-dried, and kept at −80 °C until metabolomics analyses.

Freeze-dried leaf sample powder (50 mg) was extracted in extract solvent (1 mL, acetonitrile: methanol: water (2:2:1), containing 0.1 mg/L lidocaine as an internal standard). The homogenate and sonicate circle were repeated three times, followed by incubation at −20 °C for 1 h. After 12,000 rpm centrifugation at 4 °C and filtering, metabolite profiling was conducted using a UHPLC system (1290, Agilent Technologies, Santa Clara, California, USA) with a UPLC HSS T3 column coupled to Q Exactive (Orbitrap MS, Thermo, Waltham, MA, USA). The mobile phase A was 0.1% formic acid in water for positive, and 5 mmol/L ammonium acetate in water for negative, and the mobile phase B was acetonitrile. The elution gradient was set as follows: 0 min, 1% B; 1 min, 1% B; 8 min, 99% B; 10 min, 99% B; 10.1 min, 1% B; 12 min, 1% B.

For metabolomics analysis, OSI-SMMS (version 1.0, Dalian Chem Data Solution Information Technology Co. Ltd., Dalian, China) was used for peak annotation after data processing with an in-house MS/MS database. The metabolites were mapped to the Kyoto Encyclopedia of Genes and Genomics (KEGG) metabolic pathways [58]. The metabolites involved in a range of central metabolic processes were identified, including glycolysis (ko00010), TCA cycle (ko00020), pentose phosphate pathway (ko00030), and carbon fixation in photosynthetic organisms (ko00710). These identified metabolites were submitted to hierarchical clustering analyzed using TBtools software (Version 1.068) [59].

### 4.5. Transcriptomics Analysis

As with the samples used for metabolomics analysis, 12 freeze-dried leave samples (four temporal leaf-developmental stages with three biological replicates) were used for transcriptomics analysis. Total RNA was extracted from each sample using a Trizol reagent kit (Invitrogen, Carlsbad, CA, USA). RNA quality was assessed on an Agilent 2100 Bioanalyzer (Agilent Technologies, Palo Alto, CA, USA) and checked using RNase-free agarose gel electrophoresis. After total RNA was extracted, eukaryotic mRNA was enriched by Oligo(dT) beads, while prokaryotic mRNA was enriched by removing rRNA by Ribo-ZeroTM Magnetic Kit (Epicentre, Madison, WI, USA). Then the enriched mRNA was fragmented into short fragments using fragmentation buffer and reverse transcripted into cDNA with random primers. After RNA quality assessment, mRNA purification, and fragmentation, 12 cDNA libraries were constructed, and raw data were obtained by Illumina paired-end sequencing using Illumina HiSeq2500 platform, which were submitted to the NCBI BioProject database under project number PRJNA657336. Paired-end clean reads were mapped to the ginkgo’s reference genome (http://gigadb.org/dataset/100613, accessed on 1 December 2019) using HISAT2. 2.4 with “-rna-strandness RF” and other parameters set as a default [60], followed by genes mapping to the Gene ontology (GO) [61] and KEGG databases to annotate their potential biological processes and metabolic pathways.

Fragment per kilobase of transcript per million mapped reads (FPKM) were used to normalize the original gene expression level. These FPKM data were used to reveal the relationship among samples by principal component analysis (PCA) and hierarchical clustering analyses. Subsequently, the differentially expressed genes (DEGs) between samples of different developmental stages were identified (FDR < 0.05 and |FC| > 2). The temporal expression profile of DEGs was analyzed, after which KEGG enrichment analyses were performed for each profile. The above statistical analysis was performed by the OmicShare tools (https://www.omicsmart.com, accessed on 20 May 2020).

Further, the DEGs involved in the photosynthesis light-dependent reaction and related pathways were identified based on the KEGG and GO annotation. In order to explore photosynthesis regulatory mechanisms, their 2 kb promoter sequences were assessed for co-regulated cis-regulatory elements using MEME Suite software (Version 4.9.0) (FIMO-thresh, 1 × 10^−6^) [62], which were used to identify transcription factors (TFs) that would likely target these genes. Transcriptome data were used to obtain expression profiles for these photosynthetic and TF genes. A co-expression analysis (Pearson correlation) was then performed to detect gene expression correlation networks, using the following criteria to filter the genes: *p* < 0.05, |r| > 0.80. The gene regulatory networks were generated by Cytoscape software (Version 3.7.1) [63].

## 5. Conclusions

This is the first report to evaluate the photosynthetic capacity of ginkgo leaves at different developmental stages. Here, we described variation in light response curves and photosynthetic parameters and found that the photosynthetic capacity (P_nmax_) and dark respiration rate (R_d_) gradually increased during leaf development, which might be a strategy for maintaining the primary carbon metabolism to regulate plant growth. Additionally, we observed temporal differences in the expression of photosynthesis-related genes identified in ginkgo leaf, and then constructed a network of the TFs regulating photosynthetic light reactions, including bHLH, WRKY, ARF, IDD, and TFIIIA. Our findings could improve our understanding of the physiological and molecular mechanisms involved in leaf development, and provided a theoretical basis (candidate genes) for improving the photosynthetic capacity and yield of ginkgo trees.

## Figures and Tables

**Figure 1 ijms-22-02601-f001:**
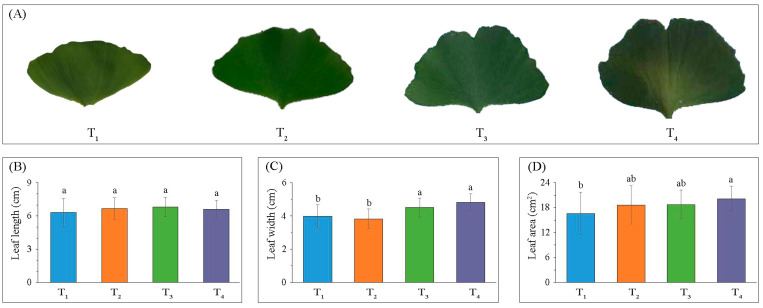
Variation in Ginkgo leaf morphology. Leaf morphology differences in Gingko leaf four development stages (**A**) with temporal differences in leaf length (**B**), leaf width (**C**), and leaf area (**D**) (error bars indicate standard deviations, and different letters represent significant differences (*p* < 0.05)).

**Figure 2 ijms-22-02601-f002:**
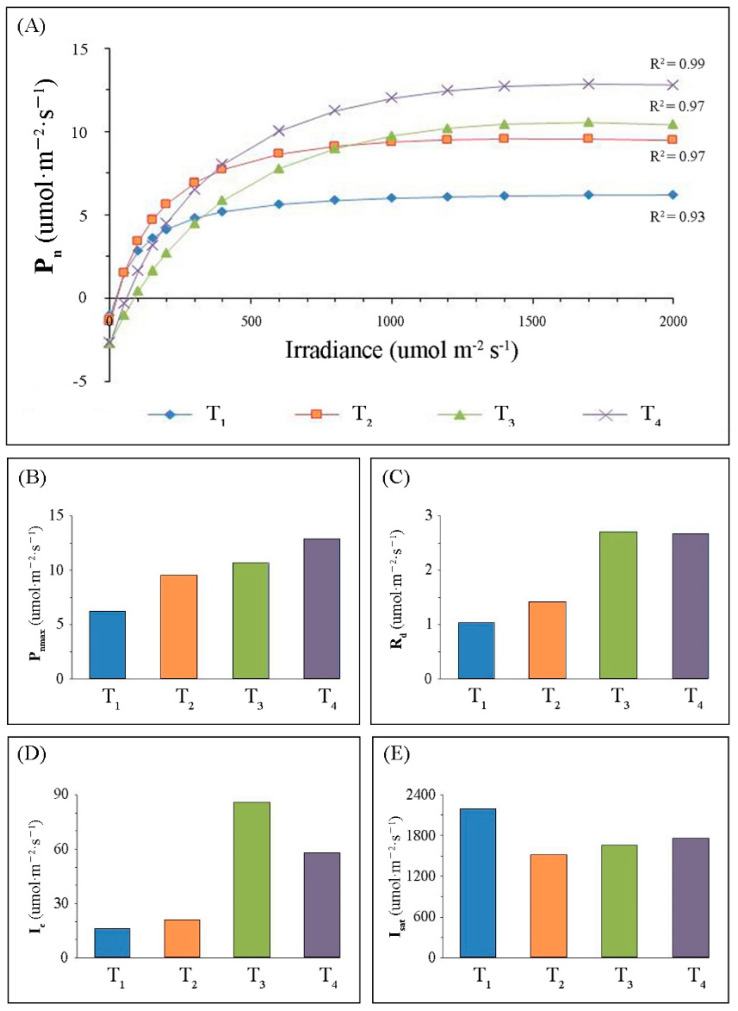
Photosynthetic light response curves (LRCs) (**A**), variation in photosynthetic capacity (P_nmax_) (**B**), dark respiration rate (R_d_) (**C**), light compensation point (I_c_) (**D**), and light saturation point (I_sat_) (**E**) over four different leaf-developmental stages (T_1_–T_4_).

**Figure 3 ijms-22-02601-f003:**
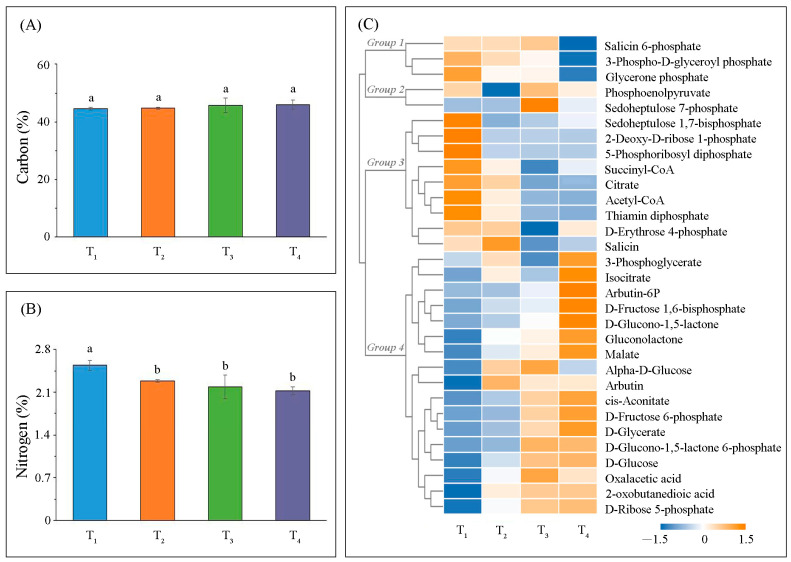
Variation in metabolic levels in the different leaf-developmental stages. Total carbon (**A**) and total nitrogen (**B**) contents (error bars indicate standard deviations, and small letters represent a significant difference (*p* < 0.05)). A hierarchical clustering heatmap highlighting quantified ginkgo metabolites presented in logarithmic scale (log_2_ (x)) where x is metabolites abundance (**C**) (change in color of the rectangle from orange to blue represents a gradual decrease in the metabolites abundance).

**Figure 4 ijms-22-02601-f004:**
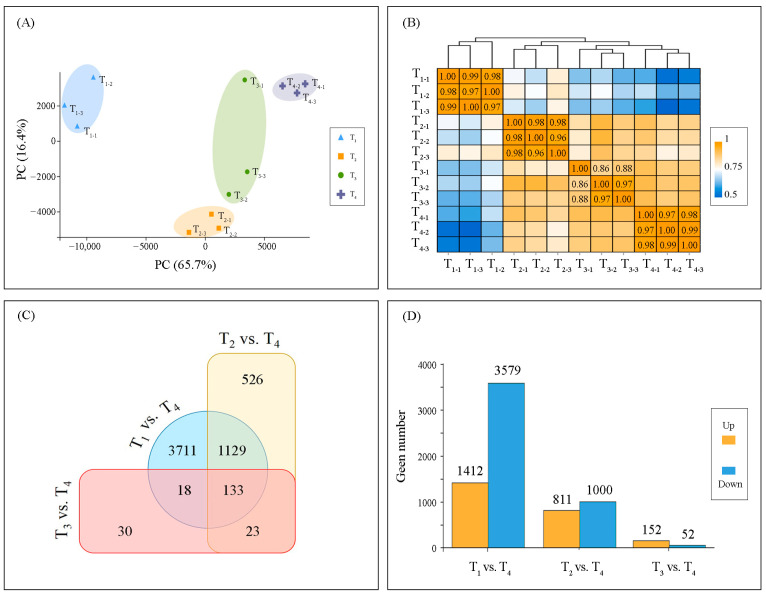
Variation in transcriptional levels in the ginkgo leaf different developmental stages (T_1_–T_4_). (**A**) PCA plots with each point representing an independent biological replicate; (**B**) Pearson product-moment correlation coefficients and clusters of the gene expression profile from leaf samples (the more orange the rectangle, the stronger the correlation between the samples, whereas the bluer the rectangle, the weaker the correlation); Taking T_4_ sample as reference, the number of differentially expressed genes (DEGs) between samples at the four development stages is depicted on Venn diagram (**C**) and column chart (**D**, yellow and blue represent up- and down-gene expression, respectively).

**Figure 5 ijms-22-02601-f005:**
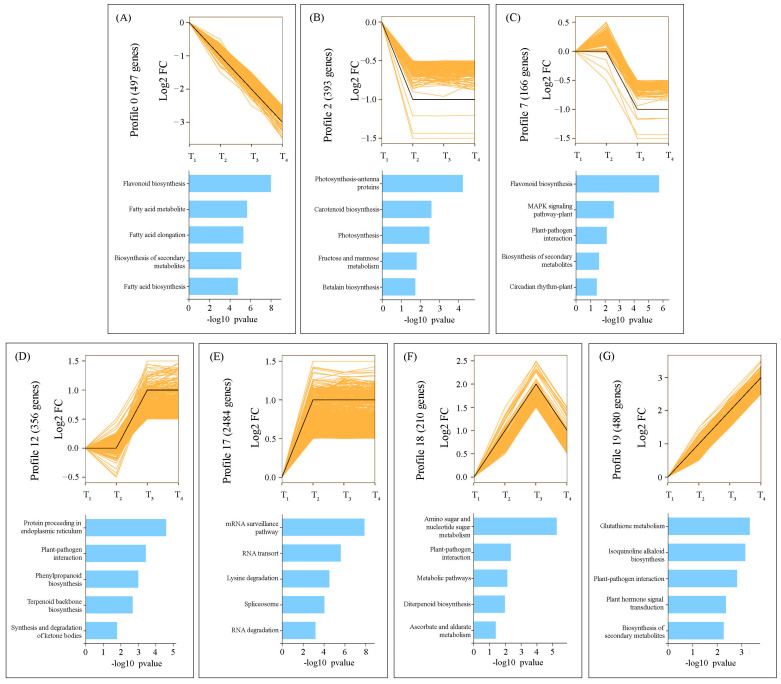
The dynamic expression patterns of differentially expressed genes (DEGs), which were grouped into seven significant clusters (*p* < 0.05) based on the similarity of their abundance profiles (**A**–**G**). Below each cluster, the top five most significantly enriched KEGG pathways are graphically represented according to their adjusted *p*-values.

**Figure 6 ijms-22-02601-f006:**
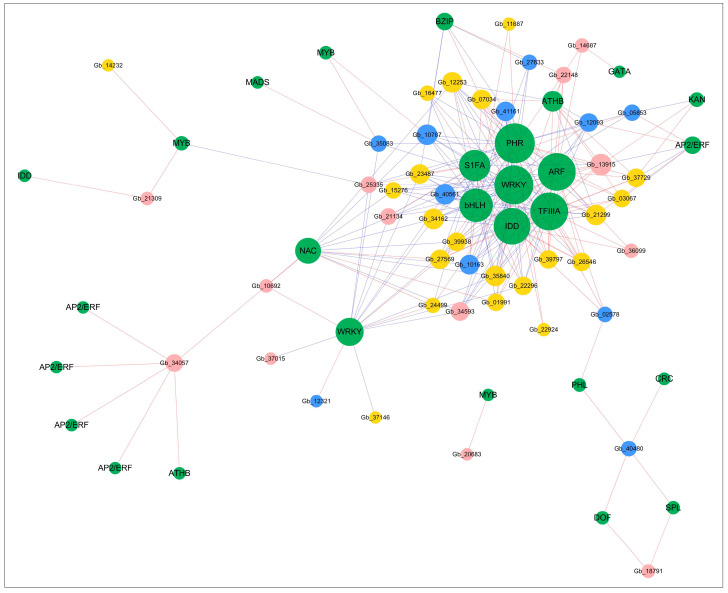
Co-expression network between TFs (green circles) and photosynthetic genes. According to genes function, the photosynthetic genes were grouped into three categories: membrane-based photosystem components (blue circles), electron transport genes (red circles), and photosynthesis regulating genes (yellow circles) (circle size is positively correlated with the connectivity of genes in the regulatory network and the red and blue lines indicate positive and negative correlations between genes (*p* < 0.05, |r| > 0.8)).

**Figure 7 ijms-22-02601-f007:**
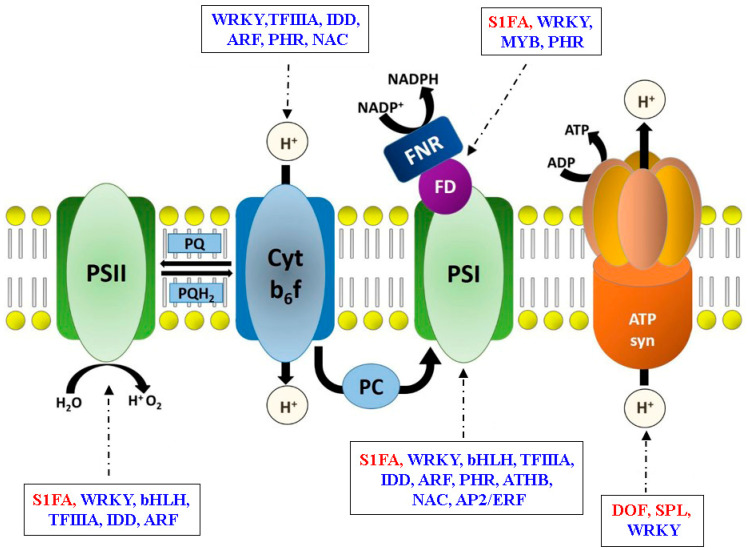
Schematic model of genes participating in the photosynthesis pathway (ko00195). The TFs were identified by the gene co-expression network. Arrows represent the regulatory relationship between transcription factors and genes encoding subunit enzymes involved in plant photosynthesis. Red and blue colors within boxes identify positive and negative regulation, respectively.

## Data Availability

https://www.ncbi.nlm.nih.gov/bioproject/PRJNA657336 accessed on 20 May 2020.

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
