# Peer review of "Metabolome and Transcriptome Analyses Reveal the Regulatory Mechanisms of Photosynthesis in Developing Ginkgo biloba Leaves"

_ijms, 2021, doi:10.3390/ijms22052601_

Round 1

Reviewer 1 Report

Metabolome and transcriptome analyses reveal the regulatory mechanisms of photosynthesis in developing Ginkgo biloba leaves

The manuscript describes variations in light response curves and several photosynthetic parameters during different developmental stages in Ginko leaves. The investigation is very interesting, giving the new insight into functioning of photosynthetic reactions during the leaf development and suggested strategy for maintaining efficient primary photosynthetic reactions that regulate plant growth. The manuscript provides valuable information on expression of photosynthesis-related genes that regulates photosynthetic light-dependent reactions that could be of valuable contribution for understanding physiological and molecular mechanism involved in leaf development. The aim of the study is described in details, however, the clearly stated hypothesis is missing.

However, there are few points for authors to be considered:

  1. The greatest emphasis in this manuscript is on changes in photosynthetic reactions and gene expression associated with differential developmental stages of Ginko leaves. The authors described that they sampled the leaves after majority leaves reach full expansion and it seems that stages were determined only by leaf age. I would appreciate if the developmental stages were described in more details.
  2. In the introduction (lines 32-33) please correct the phrases light reactions to light-dependent reactions and dark reaction to light-independent reaction.
  3. Although the conclusion sums the main contribution of the investigation, I suggest that authors improve it by adding couple of sentences that sums up their findings on photosynthetic performance of leaves, carbon metabolism and transcriptional regulatory networks (such as mention in lines 241-242 and 266-268). In that way it would be easier to follow their main findings.

Sincirely

Author Response

Reviewer #1:

  1. The greatest emphasis in this manuscript is on changes in photosynthetic reactions and gene expression associated with differential developmental stages of Ginko leaves. The authors described that they sampled the leaves after majority leaves reach full expansion and it seems that stages were determined only by leaf age. I would appreciate if the developmental stages were described in more details.

Response: we have moved Fig. S1 to the manuscript and was given the Fig. 1 title and this has resulted in renumbering all existing Figures.

  1. In the introduction (lines 32-33) please correct the phrases light reactionsto light-dependent reactions and dark reaction to light-independent reaction.

Response: suggestion has been considered (L:32-34).

  1. Although the conclusion sums the main contribution of the investigation, I suggest that authors improve it by adding couple of sentences that sums up their findings on photosynthetic performance of leaves, carbon metabolism and transcriptional regulatory networks (such as mention in lines 241-242 and 266-268). In that way it would be easier to follow their main findings.

Response: suggestion has been considered (L: 427-433).

Reviewer 2 Report

Dear Authors,

Your IJMS report on Gingko developing leaves was a pleasure to review. Therein, you combined morphological, physiological (photosynthesis-related), metabolomic, and transcriptomic analyses to conclude on the underlying mechanisms and illuminate the obvious research gap for that important plant species. It seems this report is the fourth in the series studying Gingko (DOIs: 10.3390/f10080705, 10.1016/j.indcrop.2020.112963, 10.3389/fgene.2020.589326) reported by your group, and it's impressive how much insights can be drawn from such an elegantly simple experimental setup. Warning: some methodological descriptions were picked by iThenticate plagiarism checker, so you might want to work on that.

In the current version of your report, I have some suggestions for minor changes. The heaviest changes are required in the materials and methods, where more details are needed. Please find my suggestions below, by the line number (or item name).

26 Gingko biloba L.: Refrain from using the same keyword in title and in keywords line.

32 dry

93 T1 to T: Please unify throughout to this (more legible and less confusing) format.

Fig. 1: Information is missing, what values are presented. Error bars are missing on all 5 panels.

119 was -> were

Fig. 2C: Please align the T1 to T4 with the corresponding heatmap columns.

122 Was P <0.05 meant to be α = 0.05? Normally post-hoc HSD would require providing this information, and the resulting letter codes will inform the significantly different or identical groups.

124 Information is missing (Fig. 2C) what scale was used for the abundance: Linear? Logarithmic? Other?

126 Comparing the -> Comparisons of

128 , it was found -> indicated

132 Tricarboxylic -> Changes in tricarboxylic

135 metabolites -> changes in metabolites

137 whose -> and their

145 different -> analyzed

160 control group -> reference ("control group" rather signifies 'no treatment' vs. 'treatment')

162 Fig. 3D: Insert - legend explanation missing.

174 proceeding -> was 'processing' meant? Or maybe 'transport'?

Fig.4: I suggest moving 'D' to the bottom row, to visually differentiate the categories between the rows.

184 cellular -> subcellular

191 ARF was not found in Fig.5. Please make sure that the nameplates match EXACTLY. Correct throughout.

Fig.5: Add legend with three factors: Color info, Size info (bubbles), Line color

Fig.6: Please add color (arrows?) to signify TF action on the genes: positive vs. negative regulation.

211 While it is well known that (if this such a widely known information, no extra introduction to this fact is necessary)

213 [25], -> [25], but

235 resynthesize -> re-synthesis

237 commonest -> most common

238 which was

241 thicker -> relatively thicker

257 Acetyl-CoA -> acetyl-CoA (unify, to match other names metabolites)

sentence 256-259: Rewrite; not clear what point this sentence tries to make.

270 sugars -> carbohydrates

272 photosynthesis -> photosynthesis-driven (or -based or similar)

277 intensely -> (maybe better: 'intensively')

283 which were

289 which -> but

289 to be

293 to regulate -> to regulate expression of (remove "expression" at the end of this sentence)

301 regulation -> regulatory

333-334: What is the average ambient PAR at the measurements' time and season, under natural lighting?

340 hyperbola -> hyperbolic

355 P <0.05 -> Do check whether α = 0.05 was intended here (as Significance level = Type I error rate).

355 Missing information: What statistical test was used for post-hoc ANOVA analyses of groups?

4.5: Details missing on RNA quality assessment, purification, fragmentation, reads cleaning. Cite Gingko's genome paper, Hisat2 (add version and parameters used), cite GO and KEGG. Information missing (l.401) how the 2-kb promoter regions were retrieved.

SUPPLEMENTARY FILE:

Table S3 (legend): factor -> factors

Figure S1: P <0.05 -> (alpha)=0.05

Figure S2: What values are presented on the graphs? What are the units used? Describe the meaning of brown vs. black lines on the graph.

Author Response

Reviewer #2:

26 Gingko biloba L.: Refrain from using the same keyword in title and in keywords line.

Response: suggestion has been considered (L: 26).

32 dry

Response: changes were made considering Reviewer #1 suggestion (L:…32-34).

93 T1 to T4: Please unify throughout to this (more legible and less confusing) format.

Response: we did not consider the recommended change as it will affect all Tables, Figures and Supplementary Files.

Fig. 1: Information is missing, what values are presented. Error bars are missing on all 5 panels.

Response: No error pars can be produced as single values were derived from the model.

119 was -> were

Response: suggestion has been considered (L: 124).

Fig. 2C: Please align the T1 to T4 with the corresponding heatmap columns.

Response: new Figure was included.

122 Was P <0.05 meant to be α = 0.05? Normally post-hoc HSD would require providing this information, and the resulting letter codes will inform the significantly different or identical groups.

Response: suggestion has been considered (L: 128).

124 Information is missing (Fig. 2C) what scale was used for the abundance: Linear? Logarithmic? Other? Response: suggestion has been considered and a new Figure was added. Scale was added (L: 129).

126 Comparing the -> Comparisons of

Response: suggestion has been considered (L: 133).

28 it was found -> indicated

Response: suggestion has been considered (L: 135).

132 Tricarboxylic -> Changes in tricarboxylic

Response: suggestion has been considered (L: 140).

135 metabolites -> changes in metabolites

Response: suggestion has been considered (L: 142).

137 whose -> and their

Response: suggestion has been considered (L: 145).

145 different -> analyzed

Response: suggestion has been considered (L: 153).

160 control group -> reference ("control group" rather signifies 'no treatment' vs. 'treatment')

Response: suggestion has been considered (L: 168).

162 Fig. 3D: Insert - legend explanation missing.

Response: suggestion was accepted and information was added (L: 170).

174 proceeding -> was 'processing' meant? Or maybe 'transport'?

Response: suggestion has been considered (L:182).

Fig.4: I suggest moving 'D' to the bottom row, to visually differentiate the categories between the rows. Response: New Figure 5 was added.

84 cellular -> subcellular

Response: suggestion has been considered (L: 191).

191 ARF was not found in Fig.5. Please make sure that the nameplates match EXACTLY. Correct throughout.

Response: we discovered a typo and a new Figure 6 was included.

Fig.5: Add legend with three factors: Color info, Size info (bubbles), Line color

Response: recommended suggestion is already included in the Figure 6 caption.

Fig.6: Please add color (arrows?) to signify TF action on the genes: positive vs. negative regulation. Response: we changed the colour of the TFs names to reflect positive and negative regulation, respectively (L: 215).

211 While it is well known that (if this such a widely known information, no extra introduction to this fact is necessary)

Response: we left it as is to introduce the relevant reference.

213 [25], -> [25], but

Response: suggestion has been considered (L: 221).

235 resynthesize -> re-synthesis

Response: suggestion has been considered (L: 243).

237 commonest -> most common

Response: suggestion has been considered (L: 246).

238 which was

Response: suggestion has been considered (L: 246).

241 thicker -> relatively thicker

Response: suggestion has been considered (L: 250).

257 Acetyl-CoA -> acetyl-CoA (unify, to match other names metabolites)

Response: suggestion has been considered (L: 265).

sentence 256-259: Rewrite; not clear what point this sentence tries to make.

Response: changes were made (L: 260-264).

270 sugars -> carbohydrates

Response: suggestion has been considered (L: 278).

272 photosynthesis -> photosynthesis-driven (or -based or similar)

Response: suggestion has been considered (L: 280).

277 intensely -> (maybe better: 'intensively')

Response: suggestion has been considered (L: 285).

283 which were

Response: suggestion has been considered (L: 297).

289 which -> but

Response: suggestion has been considered (L: 297).

289 to be

Response: suggestion has been considered (L: 297).

293 to regulate -> to regulate expression of (remove "expression" at the end of this sentence)

Response: suggestion has been considered (L: 302).

301 regulation -> regulatory

Response: suggestion has been considered (L: 309).

333-334: What is the average ambient PAR at the measurements' time and season, under natural lighting? Response: we did not have prior knowledge of the average eminent PAR as it was determined inside the apparatus.

340 hyperbola -> hyperbolic

Response: suggestion has been considered (L: 348).

355 P <0.05 -> Do check whether α = 0.05 was intended here (as Significance level = Type I error rate). Response: suggestion has been considered (L: 364).

355 Missing information: What statistical test was used for post-hoc ANOVA analyses of groups? Response: additional information were added (L: 363-364).

4.5: Details missing on RNA quality assessment, purification, fragmentation, reads cleaning. Cite Gingko's genome paper, Hisat2 (add version and parameters used), cite GO and KEGG. Information missing (l.401) how the 2-kb promoter regions were retrieved.

Response: additional materials were added (L: 393-404).

SUPPLEMENTARY FILE:

Table S3 (legend): factor -> factors

Response: suggestion has been considered (L: 5).

Figure S1: P <0.05 -> (alpha)=0.05

Response: suggestion has been considered and was incorporated in the manuscript as Figure 1.

Figure S2: What values are presented on the graphs? What are the units used? Describe the meaning of brown vs. black lines on the graph.

Response: numbers inside Figures represent the P-values and lines colors were included in the Figure legend (L: 13-14).